# Interface Characterization within a Nuclear Fuel Plate

**James Smith ***, **Clark Scott, Brad Benefiel and Barry Rabin**

Measurement Sciences, Idaho National Laboratory, Idaho Falls, ID 83402, USA; Clark.Scott@inl.gov (C.S.); Bradley.Benefiel@inl.gov (B.B.); Barry.Rabin@INL.gov (B.R.)

\* Correspondence: James.Smith@inl.gov; Tel.: +208-881-0926



**Featured Application: Quantitative characterization of the fuel to cladding interface in a nuclear fuel plate is being developed to qualify the fuel fabrication process as well as the fuel performance during irradiation.**

**Abstract:** To predict the performance of nuclear fuels and materials, irradiated fuel plates must be characterized efficiently and accurately in highly radioactive environments. The characterization must take place remotely and work in settings largely inhospitable to modern digital instrumentation. Characterization techniques based on non-contacting laser sensing methods enable remote operation in a robust manner within a hot-cell environment. Laser characterization instrumentation can offer high spatial resolution and remain effective for scanning large areas. A laser shock (LS) system is currently being developed as a post-irradiation examination (PIE) technique in the hot fuel examination facility (HFEF) at the Idaho National Laboratory (INL). The laser shock technique will characterize material properties and failure loads/mechanisms in various composite components and materials such as plate fuel and next-generation fuel forms in high radiation areas. The laser shock-technique induces large amplitude shock waves to mechanically characterize interfaces such as the fuel–clad bond. As part of the laser shock system, a laser-based ultrasonic C-scan system will be used to detect and characterize debonding caused by the application of the laser shock. The laser shock system has been used to characterize the resulting bond strength within plate fuels which have been fabricated using different fabrication processes. The results of this study will be to select the fabrication process that provides the strongest interface.

**Keywords:** laser shock; laser ultrasonic NDE; sensor; nuclear fuel plate

## 1. Introduction

The United States High Performance Research Reactor (USHPRR) Fuel Development (FD) pillar is tasked with the development and qualification of a novel high density U–Mo alloy based fuel which will enable USHPRR conversions to low-enriched uranium (LEU). The notable FD undertakings include the demonstration of novel fuel that passes the operational safety, dimensional stability, thermal stability, and other requirements of the applicable regulatory agency. The main FD project objective is to advance the technical means necessary to replace highly enriched uranium (HEU) fuel with LEU fuel in research and test reactors. The use of LEU fuel must be done without significant penalties in performance, economics, or safety of the reactors. FD is focused on a milestone to convert all designated research reactors, foreign and domestic, to LEU fuel.

Irradiated fuel testing is the only effective method to assess the behavior of new fuels. A common reason for fuel failure in a reactor is the debonding of the fuel from Al cladding. The void initiated by a debond generates a localized rise in temperature that may cause fuel failure by melting the cladding. The LS characterization technique will characterize bond strength at the fuel–clad interface in fresh

fuels and irradiated fuels. A quantitative measure of the bond strength at the fuel interfaces will permit the qualification of the LEU fuel and the understanding of its performance for use in research reactors.

The advancement of newly developed fuels for performance, safety and nuclear safe guard applications requires the means to characterize the performance of reactor plate fuels. Irradiated plate fuels must be tested in a high-rad environment. The plate fuel characterization is performed remotely and in a hostile environment to sensors and electronic instrumentation. Laser-based techniques have the capability to be remote and robust within a hot-cell environment. Laser-based characterization techniques provide valuable spatial resolution appropriate for scanning and imaging large areas. The Idaho National Laboratory (INL) is designing a laser shock characterization system that will examine irradiated fuel plates within a hot cell. The laser shock system is comprised of a laser shock subsystem and laser-based ultrasonic C-scan system (LUT). The LS and LUT subsystems will be integrated into a single system located at the same hot-cell window. The LS subsystem is designed to characterize interface strength and the LUT is designed to ultrasonically characterize/image irradiated material.

Laser-based characterization of base monolithic U-10Mo fuel is being conducted by the USHPRR Fuel Qualification project to support fabrication development and ensure the fuel maintains mechanical integrity and does not delaminate during normal operating conditions. Characterization involves laser shock experiments, combined with laser-based ultrasonic scanning, to evaluate the bonding integrity and relative strength of interfaces within fuel plates. This paper discusses fresh fuel studies that were carried out on a series of fuel plates manufactured at Los Alamos National Laboratory in which various controlled levels of interfacial contamination were intentionally introduced into the fuel plates during fabrication in order to affect the bond strength adversely. Additional scoping studies were conducted on several developmental fuel plates supplied by the USHPRR Fuel Fabrication Pillar in which the Zr diffusion barrier was applied to the U-Mo foils by electroplating and plasma spraying. The electroplating and plasma spraying processes are being evaluated as potential alternatives to the baseline co-rolling fabrication process.

## 2. Overview of the Laser Shockwave Technique (LST)

The characterization methods, equipment and techniques, advantages, limitations, etc., have been described in detail previously [1–7]. Prior to the development of the laser shock system, the program tried many mechanical testing techniques. Most "pull or peel" techniques required bonding an attachment to the plate. In most cases, the attachment mechanism failed first and since the fuel plates are thin there were concerns about altering the stress distribution in the plates. In general, the laser shock technique involves application of two complementary experimental techniques, laser-shockwave testing and laser-ultrasonic imaging, collectively referred to as the laser shockwave technique (LST), which allows the integrity, physical properties and interfacial bond strength in fuel plates to be evaluated. Characterization results include measurement of layer thicknesses, elastic properties of the constituents, and the location and nature of generated debonds.

In laser shock testing, a high-power pulsed laser is used to generate shock waves within the specimen. The shock wave travels as a compression wave through the material to the free (unconfined) back surface, and reflects back through the material as a rarefaction (tensile) wave. This tensile wave is the physical mechanism that produces internal failure (i.e., interfacial delamination) within the specimen. To elevate the efficacy of the optical-to-mechanical energy transfer [2–6], the surface of the specimen is covered with an optically absorbing black tape and then covered with a transparent constraining medium as shown in Figure 1. To keep the plasma contained and enhance energy transfer, the INL system uses a strong high-temperature transparent tape. The generated shock wave, which is produced under the confinement cover, causes high-amplitude molecular displacements on the surface of the aluminum cladding. The shock wave source size (roughly the laser spot size) needs to be about two times the sample thickness (about 1.5 mm) to approximate one-dimensional (1-D)

wave propagation. Under the 1-D approximation, shear stresses are neglected, and the shock wave is exclusively compressive when first generated.

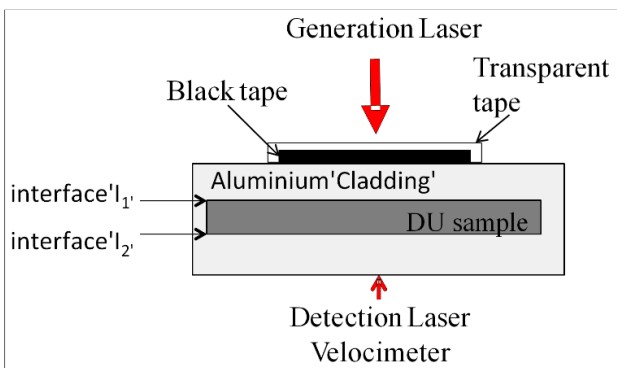

**Figure 1.** The plate fuel specimen, geometries, plasma constraining mechanism and back surface velocity detection for laser shock testing is shown.

During testing of a fuel plate specimen, bond strength is determined by increasing the laser pulse energy incrementally, which increases the amplitude of the interrogating shockwave. The threshold stress value is imposed at the interface when debonding is imminent. The stress is indirectly calculated from the measured back-surface velocity, $u$, (see Figure 1) as recorded by an optical velocimeter based on a solid Fabry–Perot etalon [8]. Assuming 1-D, elastic wave propagation, and ignoring multiple reflections, the simplified relation between the back-surface velocity and interior stress is given by Equation (1) [2–7].

$$\sigma(t) = \rho c u(t)/2 \tag{1}$$

where,

$c$ = Speed of sound in Al (6.4 mm/µs; 6400 m/s; measured)
$\rho$ = Al 6061 density (2.70 g/cm$^3$; 2,700 kg/m$^3$)
$u(t)$ = Measured surface velocity (m/s).

Equation (1) simply quantifies the maximum interior stress that makes it through the specimen to the back surface when the internal interfaces are intact. When an interface is debonded or has deteriorated, energy is dispersed or absorbed. While the shock wave energy reaching the back surface must have made it across at least partially intact internal interfaces, the actual stress field at the internal interface is more complicated, and may be higher or lower, due to specimen geometry effects and interaction of multiple reflections [2–4] from all interfaces.

To be able to identify debonding of the interfaces during a test, the LST system also contains a fully functional and independent laser-based ultrasonic (LUT) imaging subsystem. This subsystem is used to identify and image debonding caused during the laser shock test. It is capable of performing standard, non-destructive evaluation (NDE) and imaging functions—such as microstructure characterization, flaw detection and dimensional metrology in complex components via ultrasonic inspection techniques. LUT images made prior to and after the laser shock interrogation are used to confirm the presence of debonds, as well as to define the size and through-thickness location of debonds created. The LUT imaging is performed without tape or confinement and is similar to a conventional ultrasonic C-scan.

The LST system at INL has been developed for and is being used to study monolithic fuel plates consisting of U-10Mo alloy foil (typically 0.2 to 0.4 mm in thickness) having an approximately 25 µm Zr diffusion barrier coating and clad in aluminum alloy 6061 (AA6061) by a hot isostatic pressing (HIP) process [9], as shown schematically in Figure 1. In the monolithic fuel system, the integrity and interfacial strength of the bond between the fuel and the aluminum cladding is important and is, therefore, the focus of this investigation. To determine the fuel–cladding interfacial strength, the shock laser energy will be increased until debonding is observed through LUT imaging. The corresponding

back-surface velocity is recorded and used for stress determination. During a typical experiment, laser shockwave interrogation will be performed at several different locations on each fuel plate specimen. Details of the experimental procedure are presented below. There also has been interest in exploring the potential use of LST for characterizing the bond integrity and interfacial strength of the non-fueled, cladding-only (Al–Al) regions of the fuel plates. Limited success in previous attempts to create cladding–cladding debonds using LST suggested an alternative test method for cladding-cladding testing was appropriate, whereby the maximum laser power would be used initially, and if debonds were observed, the laser energy would be lowered until the debond threshold was found. Again, experimental details are presented below.

## 3. Materials and Methods

### 3.1. Variable Bond Strength Specimens

Los Alamos National Laboratory (LANL) fabricated and supplied to INL a series of fuel plate specimens in which fuel foils were fabricated by the baseline co-rolling process [9]. The specimens included both fuel–cladding and cladding–cladding fuel plate segments. The intent of this fabrication work was to produce samples having variable bond strength, using either oxide or hydrocarbon contamination introduced in a controlled manner during the fabrication process. The resulting fuel plate bonds were evaluated by the LS bond strength measurement techniques under investigation by the program to access the capability for distinguishing bond strength variations due to potential fuel production upset conditions. Upset conditions are anticipated in the production environment that may negatively influence as-fabricated bond strength. Details of the fabrication and characterization of these specimens were documented by LANL but are confidential.

Cladding–cladding testing was performed on the following specimens:

1. 2B5, 2C5—Thick oxide Al-Al bond
2. 1A5, 1B5—Thick HC Al-Al bond
3. 1D5, 1E5—Thin HC Al-Al bond

Fuel–cladding specimens supplied by LANL contained depleted uranium (DU) as the fuel. Testing was performed on the following specimens:

1. 4D5—Thick oxide Al to Zr bond
2. 3C5—Thick HC Al to Zr bond
3. 4A5—Thin HC Al to Zr bond
4. 4E5—As cleaned (thin oxide layer)

### 3.2. Alternative Zr Coating Process Specimens

The USHPRR Fuel Fabrication (FF) Pillar supplied INL with several developmental fuel plate test specimens that were used for scoping studies. All the specimens consisted of Depleted Uranium (DU) fuel foils; however, the Zr diffusion barrier was applied using alternative fabrication processes under development by FF. Most specimens contained fuel foils that were Zr coated by an electroplating process at Pacific Northwest National Laboratory (PNNL), whereas one specimen contained two partial fuel foils that were Zr coated by plasma spray at LANL. All of these fuel plates were clad in 6061 Al by a HIP process conducted at the INL [9] with nominally identical conditions. The PNNL electroplated foils were clad in HIP run 102 while the LANL plasma spray foils were clad in HIP run 104.

The following fuel plate test specimens underwent LST testing:

1. 102-C-2B (electroplated Zr, as-plated)
2. 102-D-2B (electroplated Zr, post-plating heat treatment)
3. 102-E-2B1 (electroplated Zr, as-plated, obvious Zr blisters on as-received foils)

4.    102-D-1 (electroplated Zr, as-plated)

5.    104-6 (plasma sprayed Zr, contained two partial foil pieces with unknown pedigree

Photographs of the four Zr electroplated fuel foils supplied by PNNL, as received at INL prior to HIP, are shown in Figure 2. PNNL foils 21C-2B and 21D-1 appeared relatively uniform and had consistent surface finish, while foil 21D-2B appears to have some surface discoloration on the right end. Foil 21E-2B1 had obvious Zr blisters on the left end and some discoloration on the right end. There are no known documentation or photos prior to HIP of the plasma sprayed Zr foil pieces that were supplied by LANL.

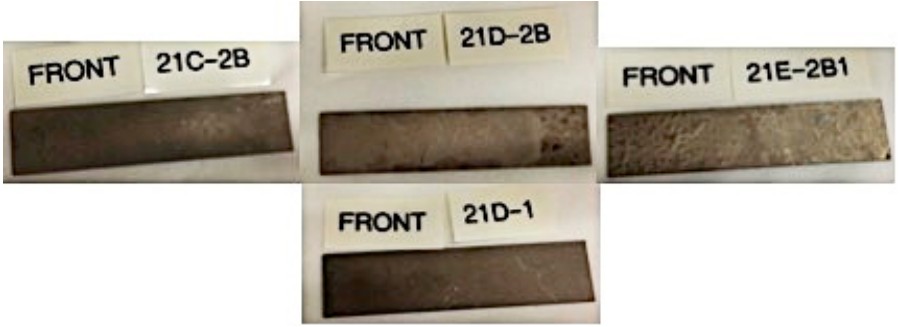

**Figure 2.** The plate fuel specimen and geometries for laser shock testing is shown.

After the plates were processed by HIP at INL, they were given a light sanding treatment but were otherwise not surface machined to a specified final thickness. Interestingly, once the foils were processed by HIP into fuel plates, the plates made from the different types of Zr coated foils appear unremarkable and show similar characteristics when imaged by conventional ultrasonic testing (UT). Figure 3 shows the through-transmission (i.e., debond) images of the fuel plates after HIP. The debond images show no signs of debonds in the as-HIP fuel plates. The debond images show the location of the fuel in the Al cladding. The fuel and cladding appear to be generally uniform and perhaps indicate some surface roughness such as on the right side of plate 104-6, which may be attributed to the sanded surface finish. It is interesting to note that, according to the UT debond image, the HIP process appears to have healed the obvious Zr blisters that were apparent on the as-received foil 21E-2B1.

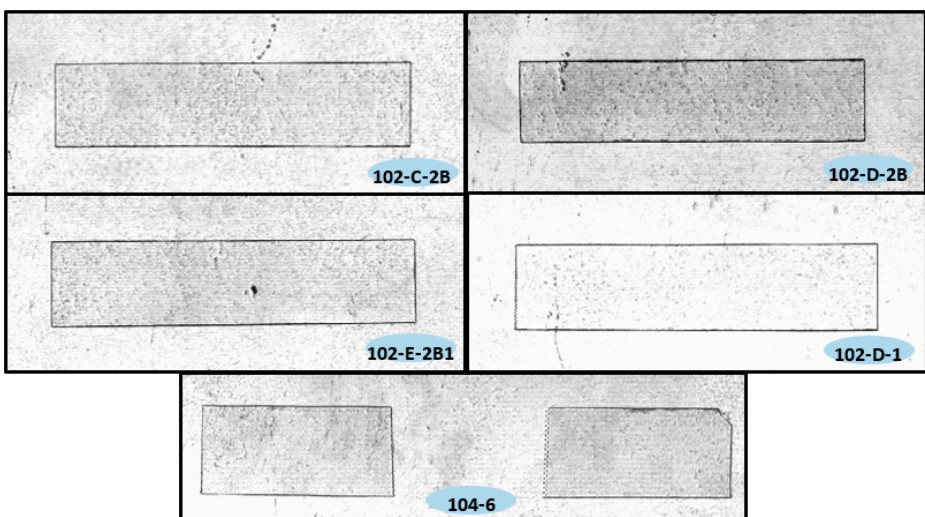

**Figure 3.** Ultrasonic C-scan images depicting bonded interfaces that let sound travel through the fuel plate. The hot isostatic pressing (HIP) process appears to have healed the coating blisters on foil 21E-2B1. Also note the presence of possible surface indentations and/or un-bonded areas indicated by the lack of transmission (dark spots).

*3.3. Test Procedures*

All testing was conducted in accordance with a test plan outlined below. For plate testing, the following containment layer and system verifications will be used. The standard sacrificial and constraining layers will be used and consist of black electrical tape (or high quality black vinyl) and 3M #8547 transparent tape, respectively [2–6]. A series of verifications and checks are performed regularly to ensure the LST system is operating correctly and providing quality data. The six-month system verification was current and the daily checklist and verification were performed each day before testing commenced.

3.3.1. Cladding–Cladding Test Steps

The test procedure for the cladding to cladding characterization is a series of process steps. The process steps are designed to ensure repeatability and enable direct comparison of the measurements between fuel plate specimens. The process steps are slightly different for the cladding to cladding characterization since the interface strength is significantly higher than fuel–cladding bonds. These steps have been closely followed.

1.  Perform cladding to cladding velocity measurements at each test location. Record and use the appropriate average cladding velocity value for the corresponding plate in the fuel-cladding tests.
2.  For the given test specimen, use test locations as shown in Figure 4 and Table 1. The lower left corner is used as a fiduciary mark.
3.  Perform ultrasonic C-scan (UT) prescan at each location to confirm the interface is intact at the start of testing.
4.  Test location #1 with laser amp delay at maximum power (40 µs ≈ 2.25 J) to determine if interface debonds.

    a.  If interface debonds, lower energy by steps of 0.25 J, using current amp-energy curve, to locate debond energy level.
    b.  If interface does not debond, test other locations at maximum power. Add two additional shocks to the last location and repeatedly shoot at maximum power to debond.

5.  Perform standard LST testing on the following matrix locations with the specimen in standard orientation, see Table 1.

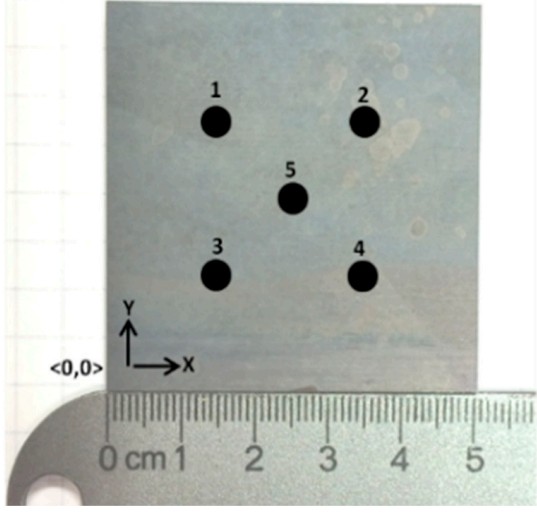

**Figure 4.** Example of clad-clad plate with locations for shock testing indicated.

**Table 1.** Example of clad–clad shock locations.

| | Al-Al Cladding Shot Points | | | | |
|---|---|---|---|---|---|
| # | 1 | 2 | 3 | 4 | 5 |
| Location x, y (mm) | 15, 35 | 35, 35 | 15, 15 | 35, 15 | 25, 25 |

The LST locations have been chosen to be toward the center of the plate to reduce edge effects. The spacing between the LST locations are spread out to keep the locations pristine. Typically, LST is not able to debond cladding-cladding interfaces.

### 3.3.2. Fuel–Cladding Test Steps

The test procedure for the cladding to fuel characterization is also a series of process steps. These process steps are slightly different from the cladding to cladding characterization since the interface strength is significantly lower for the fuel–cladding bonds. Thus far all fuel–cladding bonds that have been characterized and debonded.

1.  Use average aluminum cladding velocities from the appropriate plates in cladding–cladding testing.
2.  Use the initial shot location to define the energy range (amp delay) to test subsequent points.
3.  For the given test specimen, mark test locations with a minimum 5 mm grid spacing to maximize the number of possible testing locations with respect to the fiduciary mark F1 (lower left corner). See Figure 5 and Table 2.
4.  Each location is to be tested until interface failure by gradually increasing laser energy from the minimum value defined at the initial shot location in steps of -5 µs for the flash lamp delay (µs).
5.  Perform standard LST testing on the matrix locations in Table 2 with the specimen in standard orientation:

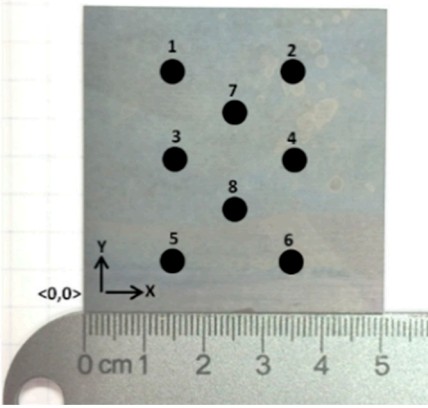

**Figure 5.** Example of fuel–cladding test fuel plate with locations for shock test indicated.

**Table 2.** Example of fuel–cladding test fuel plate shock location.

| | Fuel–Cladding Test Fuel Plate Shot Points | | | | | | | |
|---|---|---|---|---|---|---|---|---|
| # | 1 | 2 | 3 | 4 | 5 | 6 | 7 | 8 |
| Location x, y (mm) | 15, 40 | 35, 40 | 15, 25 | 35, 25 | 15, 10 | 35, 10 | 25, 32 | 25, 17 |

The LST locations have again been chosen to be toward the center of the plate but are more closely spaced. The laser energy necessary to debond fuel to cladding interfaces is significantly lower and the affected area is considerably smaller. The spacing between the LST locations can be closer together.

### 3.4. Specimen Geometry and Nomenclature

To simplify the discussions that follow, the naming conventions for referring to sample orientation and interfaces used in stress calculations are shown schematically below in Figure 6. It is important to note that the identifying aspects of the fuel plate (e.g., front surface, interface I2, back cladding etc.) are established relative to the plate identification (ID) and is independent of the orientation of the fuel plate as measurements are being made.

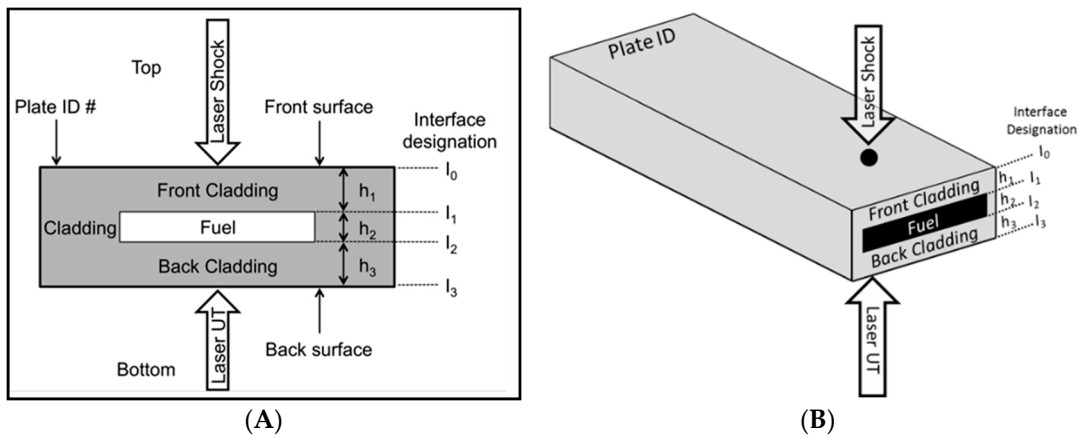

**Figure 6.** Diagrams showing the plate geometry and nomenclature used for laser shock testing. (**A**) is the cross-sectional figure, and (**B**) shows the experimental geometry.

Alternatively, in the experimental reference frame, LS is always carried out at the top (front) of the specimen and velocimeter and LUT measurements are always performed from the bottom (back) of the specimen. The reference or standard orientation is considered to be the case where the plate ID is facing up, i.e., it faces the shock laser, and the flipped orientation is where the plate ID is facing down, facing the velocimeter and laser-UT detection system.

## 4. Results and Discussion

### 4.1. Al–Al Interfaces

The testing on the Al–Al interfaces was not able to produce any debonding despite causing significant plastic deformation in the shock surface in the form of dimples. The maximum surface velocity measured was 96.4 m/s in the 1B5 plate. The maximum velocities varied from 84 to 96 m/s in the cladding-cladding test plates from LANL. The 96.4 m/s corresponds to a measured internal stress of 833 MPa using the simplified stress formula in Equation (1). Thus the Al–Al interface dynamic strength is greater than 833 MPa for all samples including the contaminated and oxide coated interfaces. Laser shock could not detect a loss in strength with either oxide or hydrocarbon contamination in the Al-Al specimens produced by LANL. This suggests that surface preparation did not cause significant changes in clad-clad bond strength.

### 4.2. Fuel–Cladding Interface in Variable Bond Strength Specimens

Thus far, all LST testing of fuel plate specimens containing dissimilar interfaces have produced debonds. This was also the case with LANL-fabricated specimens tested in this work; however, reliable testing results were only obtained after machining the fuel plates to a final thickness matching the historical nominal thickness for finished fuel plates. The as-received LANL-fabricated specimens were not initially machined to typical finished fuel plate final thickness. As-received, the specimens were 2.64 mm thick, whereas finished fuel plates are typically ~1.4 mm thick. Initial shock testing was performed on the thicker as-received specimens and it is believed that debonding was achieved. The issue was that the determination of debonding was hard to distinguish in the LUT C-scan images.

The usual debond signatures were either not present or not readily identifiable in the thicker plates. Thus the testing produced indeterminate results. A decision was made to machine the plates down to thicknesses of 1.65 mm prior to further LST testing. The LST testing was then repeated in the thinner fuel plates. The resulting ultrasonic C-scans showed the same distinct signals and signatures as observed historically during testing with the typical final fuel plate thickness [3–7].

The changes in the ultrasonic signals could have been brought about by the geometry changes and/or the work hardening by the surface machining. This remains an area for further investigation. Since the stresses at the interfaces are complex and are influenced by geometry and propagation speed of the shock wave (which are influenced by residual stresses and work hardening of the Al), there may be certain plate geometries that are not conducive to laser shock testing. All of the results reported in this section were obtained on the machined plates having a nominal thicknesses of 1.65 mm.

Figure 7 shows the resulting maximum surface velocities obtained for the LANL variable contamination specimens. The reference specimen is 4E5 with the Zr coating that was cleaned using typical procedures prior to HIPing. The contaminated samples 4D5 and 4A5 show some interface strength degradation, which might be expected. Sample 3C5 with a thick hydrocarbon layer shows a slight increase in interface strength with respect to reference specimen 4E5. These results also suggest that surface preparation has minimal effect on bond strength. Although the increase in the interface strength is not statistically significant, it is interesting to note that the bulge testing performed at LANL on specimens taken from the same fuel plates also shows the thick hydrocarbon sample having the highest interface fracture energy [10], see Figure 8. At present, the bulge test is the only known technique to compare results and help validate the laser shock technique.

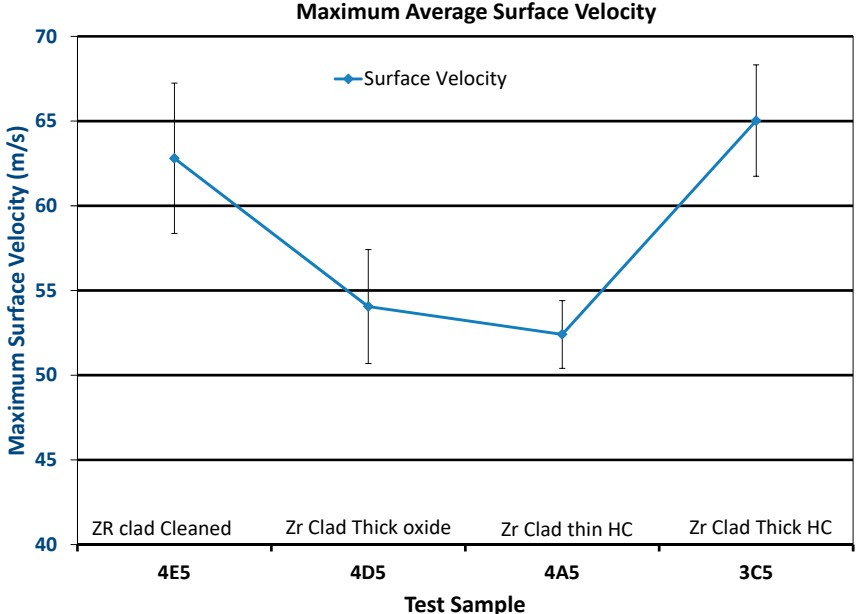

**Figure 7.** This graph shows the measured maximum surface velocity measured in the various Los Alamos National Laboratory (LANL) interface samples with varying levels of contamination.

The debond areas for the variable strength specimens are shown in Figure 9. These images are only used for a debond or bond determination. A debond has been arbitrarily defined to be two or more black pixels in the image. The size of the debond is not a concern at present. Interesting qualitative observations are made from the images For example, in most laser shock samples tested to date, the maximum debond area is on the order of the incident shock laser beam diameter, and may be smaller depending on the severity of the debonding force. As shown in Figure 9, the debond area in these specimens was typically less than the incident shock laser beam size that had a diameter of approximately 23 mm.

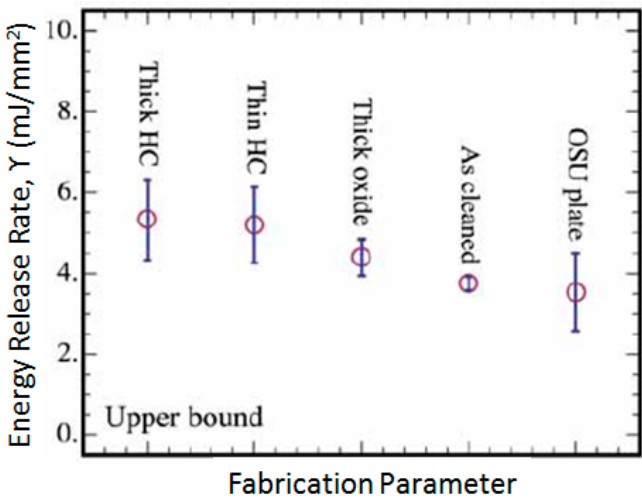

**Figure 8.** Results from bulge testing performed at LANL indicating that contaminated plates resulted in higher interfacial fracture energy compared to the plate made using the typical cleaning process [10].

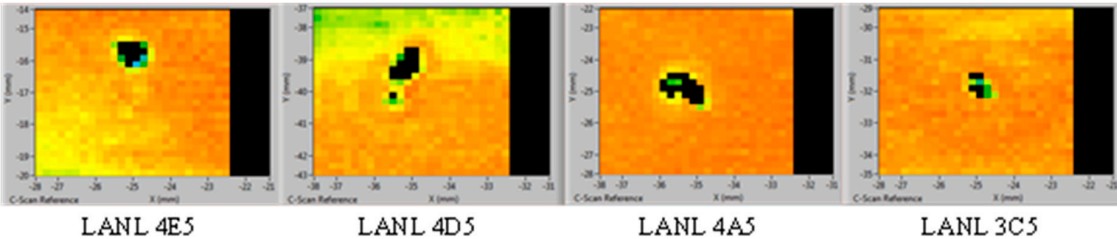

**Figure 9.** Representative images of the colorized debond locations in LANL plates caused by laser shock testing are shown. Note: The images are not to scale.

Using the simplified formula given in Equation (1), the debond threshold velocities can be converted to internal stresses and an estimate of the threshold for the interface strength can be determined as shown in Table 3. The internal stress given in the table is the highest stress obtained before the interface debonded. For comparison, the general mechanical properties for AA6061-O are the following [11]: ultimate tensile strength, 160 MPa and yield strength, 79 MPa. Keep in mind that materials can handle significantly higher dynamic stresses due to loss mechanisms such as dissipation (viscous loss) and conversion to heat.

**Table 3.** Maximum velocities are converted into estimates for the threshold interface debond strength for the variable bond strength fuel plate specimens produced by LANL.

| Specimen | Maximum Velocity (m/s) | Standard Deviation (m/s) | Internal Stress (MPa) | Deviation (MPa) |
|---|---|---|---|---|
| 4E5 | 62.8 | 4.4 | 543 | 38 |
| 4D5 | 54.0 | 3.4 | 467 | 29 |
| 4A5 | 52.4 | 2.0 | 453 | 17 |
| 3C5 | 65.0 | 3.3 | 562 | 29 |

*4.3. Fuel–Cladding Interface in Alternative Zr Coating Processes Specimens*

The plates with the electroplated and plasma Zr coatings are shown to be weaker by 200% to 300% than the co-rolled foils. Figure 10 shows that the maximum back-surface velocity for the plated and plasma coatings ranges from 15 to 30 m/s, while the co-rolled plates have a range from 53 to 65 m/s as shown in Figure 7. The lowest maximum velocities near 20 m/s are near the lower limit of where the LST system is able to make reliable measurements. The approximate debond areas for most of the alternative Zr-coated plates are generally typical and are less than the 2–3 mm incident shock laser beam diameter, as shown in Figure 11. The atypical specimen is plate 102-E-2B1 that contains the foil

that had obvious Zr coating blisters and discoloration in the as-received condition. The debond area for 102-E-2B1 has a diameter close to 8 mm, which is significantly larger than the incident shock laser spot size. As expected, plate 102-E-2B1 also exhibited the weakest fuel–cladding interface. The plasma sprayed foil in plate 104-6 also exhibited a weak interface, showing a bond strength comparable to that observed for blistered Zr coated electroplated foil in plate 102-E2B1.

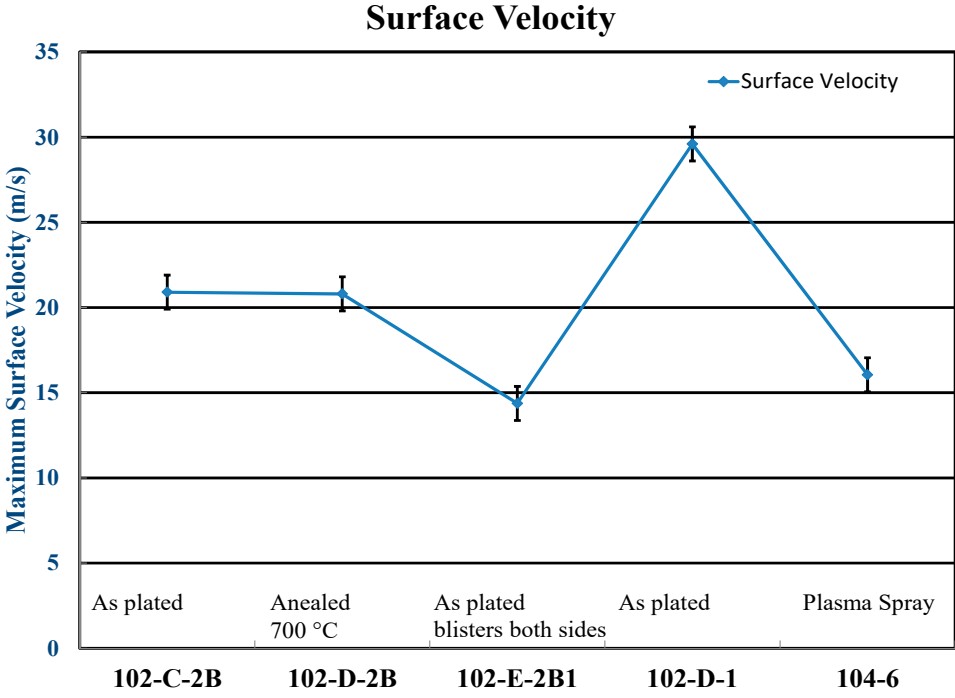

**Figure 10.** This graph shows the measured maximum surface velocity measured in the various Pacific Northwest National Laboratory (PNNL) interface samples with variable process parameters. The fuel plates contain the corresponding fuel foils: 21C-2B, 21D-2B, 21E-2B1, 21D-1.

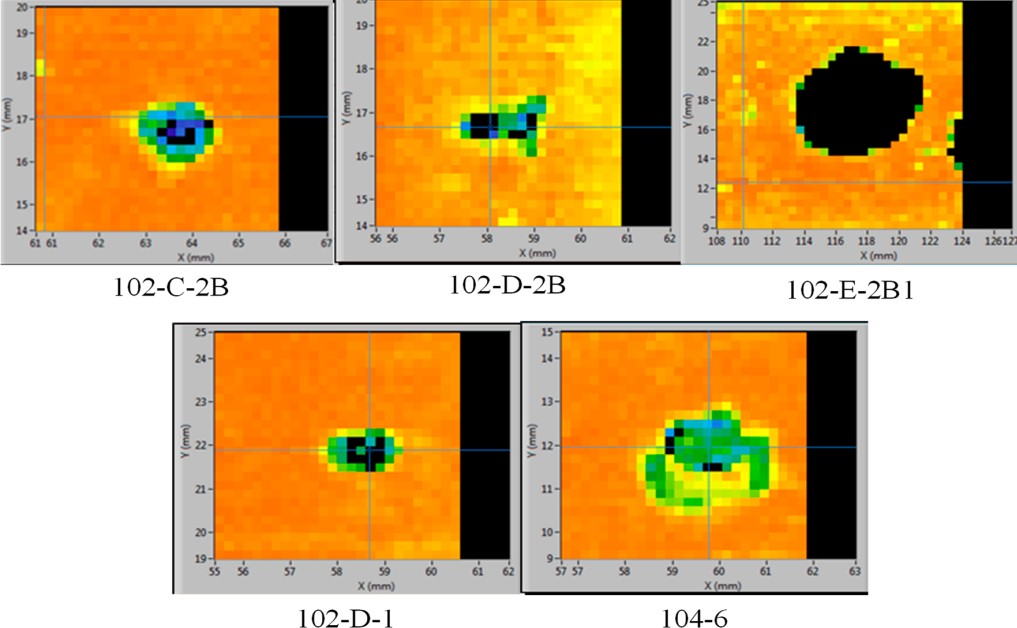

**Figure 11.** Representative images of the colorized debond locations caused by laser shock testing in PNNL plates are shown. Note: The images are not to scale.

The strongest fuel–cladding bond was observed for foil 21D-1. It is unknown what changes were made in the electroplating process for this foil if any were made. The foils did not come with documentation. Assuming that as-plated foil 21C-2B can be considered the reference condition for electroplated Zr, it would appear that annealing the foil after electroplating but prior to HIP had no significant influence on the interface strength, as shown by annealed foil 21D-2B in Figure 10. The conversion of maximum velocities to stress via Equation (1) is shown in Table 4. The debond stress for the electroplated foils are significantly lower than the co-rolled foils listed in Table 3.

**Table 4.** Maximum velocities are converted into estimates of threshold interface debond strength for the FF plates.

| PNNL Specimen | Maximum Velocity (m/s) | Standard Deviation (m/s) | Internal Stress (MPa) | Deviation (MPa) |
|---|---|---|---|---|
| 102-C-2B | 20.9 | 7.0 | 181 | 60 |
| 102-D-2B | 20.8 | 4.9 | 180 | 42 |
| 102-E-2B1 | 14.4 | 2.8 | 124 | 24 |
| 102-D-1 | 29.6 | 2.6 | 256 | 22 |
| 104-6 LANL plasma | 16.1 | 4.0 | 139 | 35 |

The debond images of the four electroplated plates and the plasma-spray plate after LST testing are shown in Figure 12. These LST plates are much more interesting to review. The dark regions indicate debonds in interfaces that reflect sound and keeps sound from traveling through the fuel plate to the other side. With the exception of plate 102-E-2B1, these debond images are typical of debonds caused by LST, being on the same order as the incident shock laser spot size. While the HIP process appears to have healed the coating blisters on foil 21E-2B1, the bond strength is low and the debond areas are large. Note that the exceptionally large debond areas for foil 21E-2B1 correspond to the blistered side of the foil (confirmed by reviewing the HIP can loading pictures); however, the interface strength appears to be similar on both halves (left and right) of the plate.

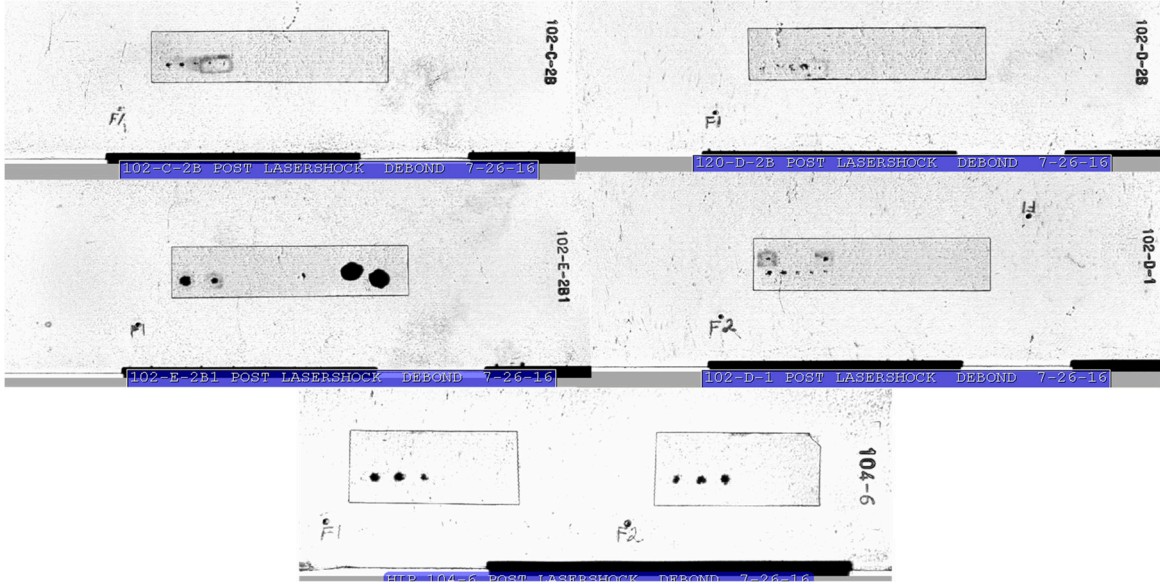

**Figure 12.** Ultrasonic C-scan images of the FF fuel plates that have been laser shock tested. The dark regions indicate debonds in interfaces that reflect sound and keeps sound from traveling through the fuel plate to the back side. While the HIP process appears to have healed the coating blisters on foil 21E-2B1, the bond strength is low and the debond areas are large. Note that the exceptionally large debond areas for foil 21E-2B1 corresponds to the blistered side of the foil.

It is also interesting to note that the plasma spray foil has a slightly higher bond strength value than the blistered foil and has a typical debond area less than 2 mm. This is an indication that there is an additional failure mechanism in the plate made with the blistered foil 21E-2B1, such that the debond size is influenced by more than just the measured bond strength. At present, it is unknown what the additional mechanism might be; however, continued crack propagation along a particularly weak or brittle interface after passage of the tensile shock wave that results in initial debonding could explain these results.

### 4.4. Microscopy on Debonded Areas of Los Alamos National Laboratory (LANL) and FF Fuel Plates

PNNL was able to perform post LST microscopy on the FF supplied 102-D-1 electroplated plate. Figure 13 shows the locations of the LST testing in the fuel plate. The debond image shows the LST locations that have been debonded as indicated by the absence of signal represented by the black circles. Figure 14 shows the micrograph of the debonded location for the debond on the lower left end of the foil. The fuel foil was sectioned through the debond vertically. One can clearly see the separation of the Zr layer from the DU fuel foil while still being attached to the Al cladding.

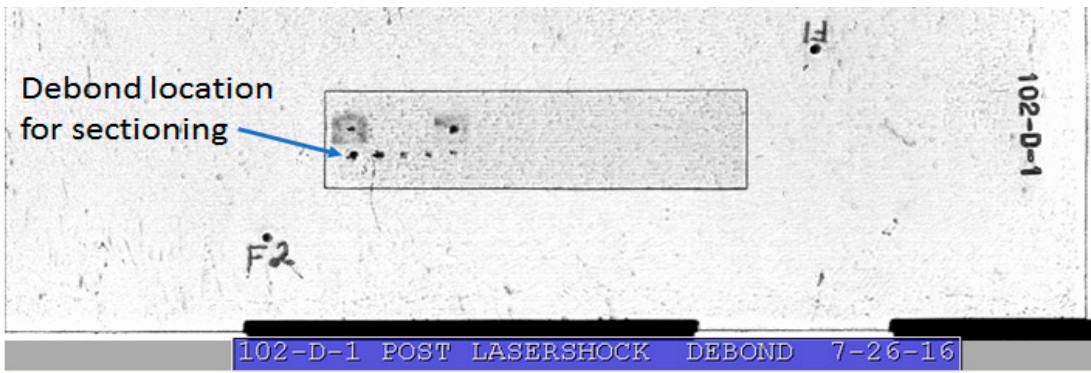

**Figure 13.** Location of the debond used for laser shockwave technique (LST) and section from an FF electroplated plasma fuel plate.

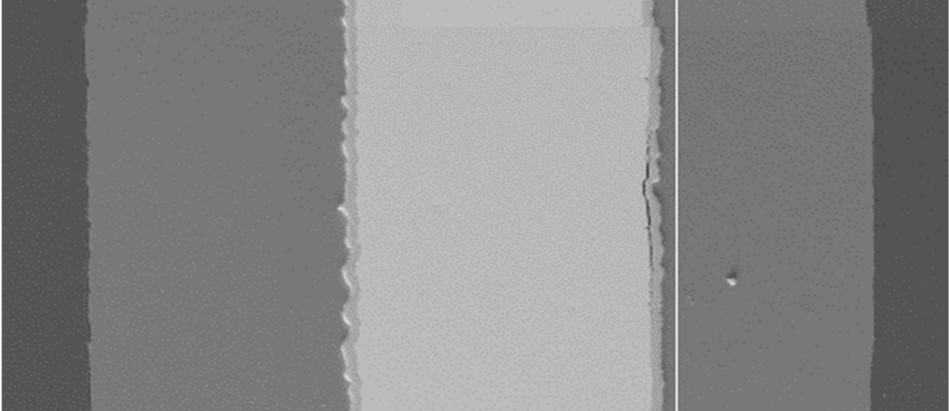

**Figure 14.** The separation of the DU fuel foil from the electroplated Zr is shown in in the right center portion of the figure. The white line is to emphasize the surface indentation which is most likely due to happenstance.

The debonding occurs on the back cladding (see Figure 6) of the plate. It is assumed that the compressional shock wave propagates first through the top cladding, followed by the foil and then the bottom cladding. Next, the shock wave hits the free back surface of the plate and reflects as a tensile wave which then pulls the interface apart on the back of the fuel foil. Figure 15 shows that the debond is on the back cladding. The "hash" between the yellow lines in the A-scan is caused by UT ringing

in the Al ligament formed by the debond. It is interesting to note that there is an indent where the debond occurred. We assume that this indentation shown in Figure 14 is happenstance since both surfaces of the AA6061 cladding do not have any indication of indentation aligned with the debond from the generation of the laser shock wave.

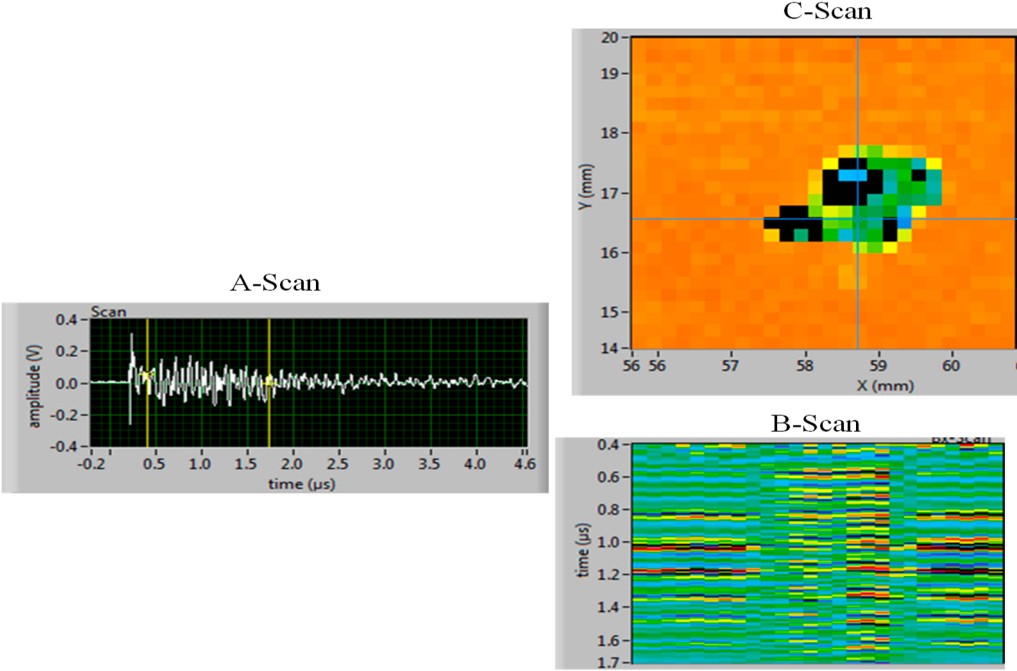

**Figure 15.** Laser UT inspection of the 102-D1 fuel plate is shown. The A-scan image shows ringing between the two yellow lines indicative of sound bouncing between a debonded ligament on the back interface (see Figure 6).

## 5. Conclusions

One of the goals of the USHPRR project is to develop a fuel plate manufacturing process that delivers fuel plates having robust and reliable fuel–cladding interfacial strength. The goal of the LST testing technique is to develop a reliable and easy-to-perform method to quantify the fuel–cladding interface strength, allowing relative comparisons to be made between fabrication variants, as well as pre- and post-irradiation. One critical step in the manufacturing process is the application of the Zr coating. As discussed in this report, bond strength testing has been performed on fuel plates manufactured at LANL using the baseline co-rolling Zr application process foils, on fuel plates with Zr-electroplated foils processed at PNNL and Zr plasma sprayed foils processed at LANL. Another significant concern relates to the cleanliness of the plate surfaces during the manufacturing of fuel plates, which was also evaluated in this study.

In addition to a nominally cleaned reference co-rolled Zr fuel plate specimen, several plates were tested having various surrogate contaminants (hydrocarbon and oxide) and different levels of contamination. The results from LST testing are somewhat surprising, suggesting that contamination introduced during manufacturing did not negatively impact the fuel–cladding bond strength. These results are corroborated by interfacial fracture energy measurements made on specimens from the same fuel plates by bulge testing at LANL. The experimental uncertainty in the data is too large to quantify differences between the different samples but it is clear that the type of surface preparation or lack thereof had little effect on the resulting interface strength.

The present results also suggest that Zr co-rolled foils have significantly stronger fuel-cladding interfaces compared to either Zr electroplated and Zr plasma sprayed foils by 200% to 300%. Another surprising result is that the fuel plate containing a Zr electroplated foil that had obvious Zr blisters in

the as-received (prior to HIP) condition exhibited a debond area at least twice the size of the incident shock laser spot size used to generate the debond, whereas typically the debond area is less than the incident shock laser spot size. The Zr plasma spray specimen has a slightly higher bond strength value than the blistered Zr-electroplated specimen and yet has a debond area less than the incident shock laser spot size, as is typical in LST testing. This is an indication that there is an additional failure mechanism in the plate made with the blistered Zr foil. Microscopy was performed on one LST location in the 102-D-1 electroplated plate. The interface separated at the Zr layer from the DU fuel foil. This is an indication that the Zr and DU interface is the weakest.

The results from laser shock testing clearly suggests that the application process of the Zr layer onto the fuel foil can make a significant difference in bond strength. The co-rolled process is seen to have higher bond strength than the electroplated and plasma sprayed process. When the LST is installed in the hot cell, similar determinations can be made on irradiated fuel to determine bond strength performance and the fabrication processes that enable high performance. The characterization of the effects due to contamination will be important to reduce fabrication costs. By understanding which types of contamination have significant effects on bond strength, the fabricators can focus their remediation efforts and practices on specific contaminates and determine the appropriate tolerance levels. It is interesting to note that test contamination did not significantly reduce bond strength as expected.

**Author Contributions:** Author contributions were as follows: Conceptualization, J.S. and B.R.; Methodology, J.S., C.S. and B.B.; Implementation, C.S. and B.B.; Writing—Original Draft preparation, J.S.; Writing—Review and Editing, J.S., C.S., B.B. and B.R.; Supervision, J.S. and B.R.

**Funding:** This publication was funded by DOE NNSA M3 (Department of Energy, National Nuclear Security Administration, Material Management and Minimization) under contract number DE-AC07-05ID14517.

**Acknowledgments:** The authors would like to thank Dave Cottle for his efforts with the laser shock testing as well as Glen Moore and his HIPing team. INL would also like to thank LANL for making and supplying documentation on the simulated contamination samples and the plasma samples. We are grateful to Cheng Liu at LANL for providing his bulge test data. We thank PNNL for suppling the electroplated samples and performing the microscopy.

**Conflicts of Interest:** The authors declare no conflict of interest.

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
