# Peer review of "Interface Characterization within a Nuclear Fuel Plate"

_applsci, doi:10.3390/app9020249_

Round 1

Reviewer 1 Report

The paper is well-written and is nearly ready for publication. There are only a few minor improvements needed. fig. 3 does not show the images well. Authors need to work on those to make sure what is planned to be shown is shown clearly.  Section 3.3.1. is just a list. A paragraph or several paragraphs are needed to open and conclude the section. Same is true for subsequent section. If authors like to focus on steps, then, perhaps, those could be shown in tables not sections. Fig. 7 - is there a reason to show values on vertical axis down to 40 m/s? The data would be shown better if the plot begins at 50 m/s. fig. 8 - what is on horizontal axis? Beyond these comments, the paper is nearly ready.

Author Response

We thank the reviewers for their time and effort in making our paper better.  Your comments are very much appreciated.

Please review the paper in NO MARKUP MODE.

Thanks 

The authors

Reviewer 2 Report

The manuscript is interesting and well written. However some issues need to be addressed before accepting it for publication, namely:

- My main suggestion is to deepen the discussion on the technological implications of the obtained results

- Additionally, I'd suggest to add further details on the validation of the used experimental method and on comparisons with alternative available ones

- Figure 9 is useless: please supply a figure with an higher resolution in order to be able to quantify the debond size

- It should be very interesting to deepen the source of the differences in behavior between foil 21D-1 and the other similar ones

Author Response

(The authors gave the same response as above.)

Reviewer 3 Report

The Laser shock subsystem and laser based ultrasonic C-scan system designed to characterize interface strength of the fuel /cladding in a nuclear fuel plate is quite interesting and has been presented in a very well documented manner. The presented results displaying the contrast of bond strength of co-rolled Zr versus electroplated or plasma sprayed is quite evident that the described procedure is sensitive enough to pick up the differences. The article should be accepted in its present form.

Author Response

We thank the reviewers for their time and effort in making our paper better.  Your comments are very much appreciated.

Please review the paper in NO MARKUP MODE.

a.      Thank you for your kind words

Thanks 

The authors